# The Tryptophan/Kynurenine Pathway: A Novel Cross-Talk between Nutritional Obesity, Bariatric Surgery and Taste of Fat

**DOI:** 10.3390/nu13041366

**Published:** 2021-04-19

**Authors:** Arnaud Bernard, Cédric Le May, Aurélie Dastugue, Audrey Ayer, Claire Blanchard, Jean-Charles Martin, Jean-Paul Pais de Barros, Pascaline Delaby, Cindy Le Bourgot, Séverine Ledoux, Philippe Besnard

**Affiliations:** 1UMR 1231 Lipides/Nutrition/Cancer INSERM/Univ Bourgogne-Franche-Comté/AgroSupDijon, 21000 Dijon, France; arnaud.bernard@u-bourgogne.fr (A.B.); aurelie.dastugue@agrosupdijon.fr (A.D.); jppais@u-bourgogne.fr (J.-P.P.d.B.); 2UMR 1087 INSERM/6291 CNRS Université de Nantes, l’Institut du Thorax, 44000 Nantes, France; Cedric.Lemay@univ-nantes.fr (C.L.M.); audrey.ayer@univ-nantes.fr (A.A.); claire.blanchard@chunantes.fr (C.B.); 3UMR 1260 INRA, 13385 Marseille, France; Jean-charles.MARTIN@univ-amu.fr; 4Lesieur Groupe Avril, 92600 Asnières sur Seine, France; pdelaby@lesieur.fr; 5Tereos, 77230 Moussy-le-Vieux, France; cindy.lebourgot@tereos.com; 6Explorations Fonctionnelles, Hôpital Louis Mourier (APHP), Colombes and Université de Paris, 92700 Nanterre, France; severine.ledoux@aphp.fr; 7Fonctions Gastro-Intestinales, Métaboliques et Physiopathologies Nutritionnelles INSERM UMR1149, CEDEX 18, 75890 Paris, France; 8Physiologie de la Nutrition, AgroSup Dijon, 26 Bd Dr Petitjean, 21000 Dijon, France

**Keywords:** diet-induced obesity, bariatric surgery, sleeve gastrectomy, fatty taste, tryptophan metabolism, kynurenine pathway, health

## Abstract

Diet-induced obesity (DIO) reduces the orosensory perception of lipids in rodents and in some humans. Although bariatric surgery partially corrects this alteration, underlying mechanisms remain poorly understood. To explore whether metabolic changes might explain this fat taste disturbance, plasma metabolome analyses, two-bottle choice tests and fungiform papillae (Fun) counting were performed in vertical sleeve gastrectomized (VSG) mice and sham-operated controls. An exploratory clinic study was also carried out in adult patients undergone a VSG. In mice, we found that (i) the VSG reduces both the plasma neurotoxic signature due to the tryptophan/kynurenine (Trp/Kyn) pathway overactivation and the failure of fat preference found in sham-operated DIO mice, (ii) the activity of Trp/Kyn pathway is negatively correlated to the density of Fun, and (iii) the pharmacological inhibition of the Kyn synthesis mimics in non-operated DIO mice the positive effects of VSG (i.e., decrease of Kyn synthesis, increase of Fun number, improvement of the fat taste perception). In humans, a reduction of the plasma Kyn level is only found in patients displaying a post-surgery improvement of their fat taste sensitivity. Altogether these data provide a plausible metabolic explanation to the degradation of the orosensory lipid perception observed in obesity.

## 1. Introduction

Recent studies suggest that the sense of taste might be compromised in obesity contributing to unhealthy food choices [1]. By providing critical information about the nature and quality of foods, gustation contributes to the decision to eat. This taste-guided ingestive behavior requires the chemo-detection of tastants by gustatory papillae mainly found in the lingual epithelium, followed by the integration of generated taste signals by specific brain structures including areas responsible for the food reward [2]. In addition to the primary tastes (i.e., sweet, bitter, salty, sour and umami), a growing number of studies pleads in favor of a sixth taste modality responsible for the perception of dietary lipids [3,4,5]. Interestingly, the orosensory perception of lipids has been also found to be weakened in obesity [5]. Indeed, diet-induced obese (DIO) rodents are unable to perceive properly low concentrations of oily solutions, obesogenic diets increasing the oral detection threshold for lipids [6,7]. Likewise, rats hypo-responsive to fatty acid stimuli preferentially consume high fat diet (HFD), likely to reach a hedonic satisfaction, and become obese [8]. The dysfunction of both the lipid signaling cascade in taste bud cells [9] and the reward pathway [10] might explain this obesity-mediated changes in the detection/perception of dietary lipids. In obese patients, the association between hyposensitivity to lipids and preferential consumption of fat-rich foods was also reported [11], although the strict relationship between body mass index and oral lipid sensitivity remains a matter of debate in human [12].

Bariatric surgeries are efficient procedures used to induce long-term weight loss and to improve associated comorbidities frequently found in obese patients [13]. Vertical sleeve gastrectomy (VSG) is often the first bariatric procedure proposed to obese patients and the most commonly performed procedures worldwide [14]. In some operated patients, a better taste sensitivity to lipid stimuli eliciting healthier food choices is reported after surgery [15,16]. However, the origin for these improvements remains elusive. Interestingly, animal studies reproduced this paradigm opening an opportunity to identify the underlying molecular mechanisms. In DIO rats, the decreased licking response to oily stimuli is widely improved after Roux-en-Y gastric bypass (RYGB) [17], and formerly obese rats preferentially consumed low-energy dense foods during multiple-choice food tests [18,19,20]. VSG rats also decrease their preference for fat-rich foods independently of the restriction of gastric volume and caloric intake suggesting that this surgical procedure modifies the eating behavior beyond its anorexic impact [21]. An alteration of the central sensory signal perception leading to a reduction of rewarding properties of palatable foods likely contributes to these VSG-mediated behavioral changes [18,21,22]. Whether obesity surgeries can also affect the function of peripheral gustatory system is unknown. Interestingly, a decreased density of fungiform papillae (Fun) was recently reported in the DIO mice [23]. Moreover, a reduction of the Fun number was also observed in obese patients displaying a degraded orosensory perception for fat stimuli and a high rating for energy-dense foods [24]. To date, the underlying mechanisms remain elusive.

Collectively these data show that the obesity state plays a role in the efficiency of the orosensory perception of lipids and raise a basic question: how do obesity and bariatric surgeries interfere with the fatty taste detection? To explore this issue without preconception, a global analysis of plasma metabolites was performed in combination with behavioral preference tests and Fun counting in both DIO mice after VSG or sham-operation (Sham). A human exploratory study was also undertaken two weeks and six months after a vertical sleeve gastrectomy.

## 2. Materials and Methods

### 2.1. Mice

These studies were conducted in the strict accordance with the European guidelines for the care and use of laboratory animals and the protocols were approved by the French National Animal Ethics Committee (APAFIS#12728-201712151028893 and APAFIS#19877-2019032109244847). Six-week-old C57Bl/6 male mice were purchased from Charles River Laboratories (France). They were individually housed in a controlled environment (constant temperature and humidity, dark period from 7 p.m. to 7 a.m.) and allowed free access to tap water and chow. Mice were fed either a standard chow (4RF21, 315.0 Kcal/100 g) or an obesogenic high fat diet (HFD—4RF21, Mucedola SRL, Milano, italia + 31.8% palm oil, wt/wt, 505.4 Kcal/100 g) [25] for two months (Figure 1A). The quantification of the fat mass was done by molecular resonance (EchoMRI—Echo Medical Systems, Houston, TX, USA). Two complementary experiments were performed using two different sets of mice.

Firstly, to explore the molecular mechanisms by which bariatric surgery might affect the fat taste, a non a priori investigation (i.e., plasma metabolomic analysis) was undertaken after VSG in obese mice (VSG-DIO, *n* = 11) and in lean and obese controls after 9 weeks under HFD (Sham-L, *n* = 14, and Sham-DIO, *n* = 15). After surgery, DIO mice were maintained on HFD until sacrifice.

Secondly, to determine the putative role played by the Trp/Kyn pathway on the orosensory perception of lipids, a pharmacological inhibition of the first rate-limiting enzyme of the Kyn pathway (i.e., indolamine 2,3-dioxygenase—IDO-1) was induced in non-operated DIO mice (*n* = 15) by addition of 1-methyl tryptophan (1-MT, 2mg/mL) in the drinking water according to [26] for 8 weeks, controls being untreated DIO animals (*n* = 15).

### 2.2. Surgical Procedures

The VSG surgical method has been fully described and video-captured previously [27]. Briefly, after a laparotomy, the stomach was externalized and 8.0 prolene sutures (Ethicon^®^, Johnson & Johnson, New Brunswick, NY, USA) were used to suture the pyloric vessels along the greater stomach curve. The gastrostomy was performed on the anatomical line from the pyloric region to the cardiac part of the stomach. The incision site was closed with 8.0 Prolene by using a running suture from the gastro-esophageal junction. Eighty percents of the stomach were removed. A special care has been taken to avoid any damage to neural and vascular systems especially at the level of esophagogastric area. The abdominal skin and muscle were closed with 5.0 Prolene interrupted sutures. The controls were sham-operated.

### 2.3. Oral Glucose Tolerance Test

To assess the metabolic efficiency of surgery, oral glucose tolerance tests (OGTT) were performed before and after surgery [28] (Figure 1B). Mice that were fasted for 6 h received an oral bolus of D-glucose (2 g/kg BW), and the plasma glucose levels were measured at 0, 15, 30, 60, and 120 min after gavage by tail bleeding (Glucometer-One Touch Verio).

### 2.4. Behavioral Test

Two-bottle choice tests were performed for 12h at the beginning of the dark period on individually housed mice according to previously published procedure [29]. In brief, the food was removed during the experiment. The mice were subjected to a choice between two bottles containing either a control solution (0.3% xanthan gum in water) and an experimental one (2% rapeseed oil in control solution). At the end of the test, the fluid intake was measured for each bottle and the preference (i.e., the ratio between experimental solution consumption and total intake) was calculated.

### 2.5. Evaluation of the Fungiform Number

After sacrificing isoflurane-anaesthetized mice by cervical dislocation, the tongues were removed, washed in 0.09% NaCl and stored in 10% neutral buffered formalin at 4 °C for 2 days. To determine the fungiform papillae density, the tongues were stained for 20 min with a methylene blue solution (0.5% in PBS, wt/v) and then washed in PBS before observation. A picture of the anterior dorsal tongue was captured using a light macroscope equipped with a camera (Axio Zoom, Zeiss, Lena, Germany), and then a 20mm^2^ area was analyzed to standardize the Fun counting.

### 2.6. Plasma Metabolomic Analysis

Metabolomics analyses were performed by Profilomic SA (Huningue, France) on plasma from blood collected at the end of the first experiment (i.e., 9 weeks after surgery—Figure 1). In brief, 50 μL of plasma were homogenized in 200 μL methanol cooled at −20 °C. The samples were vortexed for 10 s and then incubated on ice for 90 min. After centrifugation (20,000 g, 10 min, 4 °C), precipitates were dried using a stream of nitrogen and then solubilized in 150 μL ammonium carbonate (10 mM, pH 10.5/can 40:60, *v*/*v*), before storage at −80 °C until the LC-MS analysis. The metabolomic profile of the samples was determined using a high-performance liquid chromatograph adapted to polar compounds (SeQuant ZIC-HILIC, Merck, Germany) coupled to a high-resolution mass spectrometer (Q-Exactive Plus, Thermo Fisher Scientific, Illkirsh, France). The mass spectra were collected using a high resolving power (70,000 Full Width at Half Maximum, FWHM) by alternating positive and negative ionization modes. The identification of metabolites in humans and mice was performed using their respective mass and retention time according to the Cribiom and Profilomic databases, respectively.

### 2.7. Plasma LC-MS/MS Analysis and LPS Determination

The extraction procedure was performed as previously described [30]. A standard internal mix (Si-Mix) containing Trp-d5 (10 pmol/µL) was prepared in acidified mobile phase (0.2% FA/0.05%, TFA/1% ACN in water). Calibration curves were performed using standards (i.e., 150–9600 pmol L-Trp, 1–64 pmol L-Kyn or 0.1–6.4 pmol KynA, 0.5–32 pmol Quinolinic acid) in 20 µL of a pool of plasmas. The samples (20 µL plasma) and calibration standards (20 µL) were mixed with 10 µL Si-Mix solution, 10 µL acidified mobile phase and 150 µL ice-cold methanol. The mixtures were maintained at −20 °C for 30 min. After centrifugation (10 min, 15,000 g, +4 °C) the supernatants were collected and evaporated to dryness under vacuum. Dried extracts were finally dissolved in acidified mobile phase (30 µL) and centrifuged (5 min, 15,000 g, +4 °C). Two microliters of the supernatant were injected into a 1260 LC system coupled to a 6460-QqQ MS/MS system equipped with an ESI-Jet Stream source (Agilent technologies). Separation was achieved on a Poroshell 120 EC-C18 2.1 × 100 mm, 2.7 µm column (Agilent technologies). The plasma LPS levels were determined by direct quantification of total 3-hydroxymyristate using 3-hydroxy-tridecanoic acid (4 pmol) as the internal standard by high-performance liquid chromatography coupled with mass spectrometry (LC-MS/MS) as previously described [31]. It is noteworthy that QA was failed to be detected in plasma samples due to a problem of standard.

### 2.8. Patients

The exploratory human study was realized in patients from the HumanFATaste2 cohort which is described elsewhere [32]. This study was approved by French ethics committees (Comité de Protection des Personnes, CPP n° 15–032 and Agence Nationale de la Sécurité des Médicaments, ANSM n° 150811B-21) and registered at Clinical Trials (NCT#02497274). Severe obese adult patients (BMI: 43 ± 4 kg/m^2^; age: 36.6 ± 1.3 y, *n* = 32) who were candidates for a VSG were selected according to the following exclusion criteria: treatment able to modify taste perception, surgical care of obesity already performed, buccal withering, diabetes (use of hypoglycemic drugs or fasted plasma glucose level >7 mmol/L), smoking habits, chronic inflammatory pathologies.

All subjects received detailed information about the study and provided a written consent. To avoid gender-mediated bias [33], this study was conducted on women only. VSG interventions were performed laparoscopically performed in the Louis Mourier Hospital (Assistance Publique Hôpitaux de Paris—APHP), as previously described [34].

Two complementary investigations were performed 2 weeks before and 6 months after VSG, in the morning, in overnight fasted patients. Firstly, a targeted plasma metabolite analysis was realized by LC-MS/MS. Plasma C-reactive protein (CRP) levels used as inflammatory marker were assayed using human CRP Elisa kit (Enzo Life Sciences, Villeurbanne, France). Secondly, the oral lipid perception threshold was determined using the 3-ascending force-choice (3-AFC) test according to a previously published procedure [35]. In brief, subjects had to identify among three samples the one that was different from the two others. The batches of samples were presented in arising concentration from 0.00028% to 5% linoleic acid (LA- wt/wt) spaced by 0.25 log units (18 concentrations in total). The orosensory perception threshold was reached when the lipid sample was correctly identified 3 times consecutively [32].

### 2.9. Statistical Analysis

For the metabolomic analysis, a partial-least-squares (PLS) analysis and multiblock hierarchical PLS were performed with SIMCA P + 12 (Sartorius, Aubagne, France) as described [36,37]. ANOVA cross-validation, permutation tests and cross-validation were used to validate all PLS models. The annotated metabolites were assigned to a biological role using the HMDB MetaboCard, Wikipedia description, PubChem description and KEGG pathways. A grouping was the done according to their functional role and then a hierarchical PLS was performed. Each biological function was scored by the hierarchical PLS SIMCA algorithm [36]. The pathway blocks were “weighted” to account for the number of metabolites per block. A PLS discriminant analysis (PLS-DA) was also performed on autoscaled data of the individual metabolites. Significantly dysregulated metabolites were determined using the variable importance in projection (VIP) parameter of the SIMCA software [36]. The univariate ANOVA and biomarker searching using receiver operating characteristics (ROC) curves were performed using MetaboAnalyst [38]. For other data, a statistical analysis was performed using the R software (3.1.2) with an alpha level of 0.05 and GraphPad Prism (Graphpad software, Inc., San Diego, CA, USA) for correlations. Normality and variance homoscedasticity were checked using the Shapiro and Fisher tests, respectively. Depending on those results, the data were analyzed with either a Student’s t test or Wilcoxon-Mann–Whitney.

## 3. Results

### 3.1. VSG Reduces the Metabolic Dysfunctions Found in DIO Mice

Oral glucose tolerance tests (OGTT) were performed 2 days before and 2 weeks after surgery (Figure 1A). VSG significantly reduced the DIO-induced glucose intolerance observed in obese animals before surgery and in sham-DIO mice after surgery (Figure 1B). These metabolic improvements correlated with the evolution of body mass, which increased in the sham-DIO group but decreased in the VSG-DIO mice despite still being fed with the HFD (Figure 1C). The beneficial effects of the VSG were persistent since fat mass, blood glucose, plasma cholesterol and LPS levels were always close to the Sham-L values at the time of sacrifice (Figure 1D). 

### 3.2. VSG Corrects the DIO-Induced Adverse Plasma Metabolite Signature

To generate new insights into how nutritional obesity disturbs the orosensory perception of dietary lipids, an untargeted analysis of plasma metabolites was undertaken. Nine metabolites were identified as discriminant the VSG-DIO mice from Sham-DIO controls (Figure 2A). The area analysis of ROC curves was used to evaluate their performance as biomarkers [39]. Values between 0.97 and 0.88 were found for all identified metabolites indicating that the differences between VSG-DIO and Sham-DIO mice are highly significative. The principal component analysis (PCA) of biomarker data confirmed that they were sufficient to clearly discriminate the VSG-DIO mice from the Sham-DIO controls (Figure 2B). The impacted pathways mainly concern the neuronal function and protein metabolism (Figure 2C). Several identified metabolites are known to play a role in neurotoxicity (quinolinic acid, QA [40], guanido-succinic acid [41]) and in neuron survival and genesis (cytosine [42], deoxy-cytosine [43]), while others directly or indirectly participate to the tryptophan (Trp) catabolism through the muscle breakdown (methyl-histidine, mHis [44]) and oxidative stress (N-acetyl-aspartic acid, AAA [45] and sebacic acid, SebA [46]) (Figure 2D). It is noteworthy that QA, is the only metabolite common to both pathways (Figure 2D). Plasma QA level is a marker of the indolamine 2–3 dioxygenase activity (IDO-1), the rate-limiting enzyme responsible for the tryptophan (Trp) degradation along the kynurenine pathway (Kyn) in inflammatory conditions (Figure 2E). Overproduction of QA leads to cytotoxic mechanisms including cell death promotion and behavioral alterations [47].

Interestingly, VSG broadly corrected the altered plasma metabolite signature found in Sham-DIO mice despite that mice were maintained on HFD. The lower plasma mHis, AAA and sebA levels observed in VSG-DIO mice (Figure 2D) suggest a reduction of both muscle breakdown and oxidative stress (Figure 2E). Consistent with these changes, the plasma QA levels were found to be significantly lower in VSG-DIO mice than in Sham-DIO animals, suggesting that VGS is associated with a decreased IDO-1 activity likely due to a reduction of the inflammatory environment (Figure 2E).

### 3.3. Density of Fungiform Papillae Is Negatively Correlated to the Kyn Pathway Activity

To delineate the phenotypic difference between VSG-DIO and Sham-DIO mice, we first performed behavioral tests. We found that the preference for oily solution was lower in Sham-DIO than in Sham-L mice. Interestingly, this phenomenon was partially corrected in VSG-DIO animals suggesting a deleterious effect of obesity beyond the HFD (Figure 3A). Long-term (12 h) two-bottle choice tests used herein bring a global information on the acceptance of a taste solution resulting from a combination of peripheral, central and post-ingestive cues.

To further explore whether a change in the peripheral detection system might explain the improvement of fat preference found in the VSG mice, an analysis of the Fun density in the anterior dorsal tongue was undertaken. Consistent to previous published data [48], a reduction of the Fun abundance was found in Sham-DIO mice as compared to Sham-L animals (Figure 3B). The VSG partially corrected this disorder. Indeed, the Fun number found in VSG-DIO mice was greater than in the Sham-DIO group but without reaching the controls’ density (Figure 3B). Interestingly, Fun number and preference for oily solution were found to be positively correlated (Figure 3C). This observation is reminiscent to what was found for sweet taste in human [18].

Sensory innervation is essential for the gustatory papillae renewal [49]. QA in excess becoming neurotoxic, it was hypothesized that the activity of the Kyn pathway might impact the Fun density. Consistent with this assumption, the Fun number was found to be negatively correlated to plasma QA levels (Figure 3D).

### 3.4. Pharmacological Inhibition of IDO-1 Enhances the Fun Number and Improves the Orosensory Perception of Oily Solution in DIO Mice

To explore this hypothesis, the pharmacological inhibition of the Kyn pathway was performed in non-operated DIO mice. In order to reduce the Trp degradation found in DIO mice, IDO-1 activity was inhibited by addition of 1-methyl TRP (1MT) in the drinking water [50] (Figure 4A). After 8 weeks of treatment, 1MT-DIO and control (C-DIO) mice displayed a similar body mass demonstrating that 1MT did not affect the weight gain in response to the HFD (Figure 4B). The effectiveness of 1MT treatment on the inhibition of IDO-1 activity was attested by both the reduction of the plasma Kyn level and the decrease of Kyn/Trp ratio found in 1MT-DIO mice (Figure 4C). By contrast, the plasma KynA levels were increased in the treated animals (Figure 4C). Since KynA exerts a neuroprotective action (Figure 4A), this data suggests that the neurotoxic status of treated-mice was lower than in controls.

Consistent with the literature [51,52], the total hydric consumption during two-bottle preference tests was higher before treatment (i.e., in lean mice) than after treatment (i.e., in obese mice—Figure 4D). Whereas the preference for the oily solution was similar in the two groups of mice before treatment, a greater preference for lipids was found in 1MT-DIO mice as compared to the C-DIO animals (Figure 4D). The fact that 1-MT treated-mice displayed a higher number of Fun than C-DIO (Figure 4E) might contribute to this orosensory change. A lower plasma LPS level was also found in 1-MT mice (Figure 4F), suggesting a reduced systemic inflammation despite that mice were maintained on HFD.

### 3.5. The Down-Regulation of Kyn Pathway Found after VSG Is Associated with an Improvement of the Fatty Taste Sensitivity in Human

In order to explore whether the overactivation of the Kyn pathway found in DIO mice might also occur in obese humans, an exploratory study was undertaken in patients (*n* = 32) 2 weeks before and 6-months after VSG. This procedure led to a massive body weight loss (Figure 5A). Trp and Kyn plasma levels and Kyn/Trp ratio assayed by MS were found to be lower after VSG than before surgery (Figure 5B). In these patients, the comparison of LA perception thresholds before and after VSG has allowed to distinguish two subgroups: patients displaying an improvement of the fatty taste sensitivity (i.e., lower LA threshold after surgery) and unresponsive patients [32]. The plasma Kyn levels and the LA perception threshold were found to be positively associated only in the LA-improved group (Figure 5C) suggesting that in these patients a high plasma Kyn level and the degradation of fatty taste sensitivity might be correlated. They were also characterized by a greater post-surgery decrease in both the plasma levels of Kyn and of the inflammation marker CRP than in not improved patients (Figure 5D), whereas no relation was found with BMI and weight loss.

## 4. Discussion

Mechanisms by which the fat taste sensitivity is altered in DIO rodents, and conversely, is improved after bariatric surgery are not yet fully known. To gain insight into this issue, an untargeted analysis of the plasma metabolome has been performed in VSG-DIO mice and Sham-DIO controls. The main result of this study is the identification of the Trp/Kyn pathway as a potential regulator of the orosensory lipid perception in the mouse. Among the nine discriminant metabolites identified by the metabolomic analysis, QA caught our attention because this kyn pathway metabolite is a neurotoxin whose overproduction due to an inflammatory context has a deleterious impact on the NMDAR-expressing target neurons [47]. This negative effect might contribute to the deficit of fatty taste sensitivity found in Sham-DIO mice. Indeed, NMDARs expressed in gustatory neurons participate to the modulation of taste signals before their transfer to the brain [53]. Moreover, these gustatory neurons play a crucial role in the taste bud cell renewal by producing neurotrophins (e.g., BDNF) [49]. These observations raise the possibility that an over-activation of the Kyn pathway during obesity might be implicated in the lower number of Fun papillae found in DIO mice (present study and [48]), and therefore in their inability to properly detect dietary lipids (present study and [6,7]). Interestingly, these troubles appear to be widely corrected in VSG-DIO mice despite they were maintained on HFD after surgery.

The fact that the QA is a marker of the IDO-1 activity, led us to explore the impact of the pharmacological inhibition of this rate-limiting enzyme of the Kyn pathway on studied parameters. The efficiency of the inhibitory action of 1-MT on IDO-1 activity is attested by the reduction of plasma Kyn/Trp ratio and Kyn levels in treated mice [26]. As already reported elsewhere [54], the plasma levels of KynA are also increased in the 1-MT treated mice due to the activation of alternative synthesis (e.g., gut microbiota [55]). Interestingly, KynA antagonizes the neurotoxic action of QA on NMDARs (Figure 4A). This protective function might explain the greater number of Fun found in 1MT-DIO mice as compared to DIO-C and, thereby, the improvement of their lipid preference. However, these positive changes remains relatively modest suggesting that other homeostatic modifications (e.g., changes in the rewarding pathway and/or in appetite-regulating hormones [4]) might also participate to the improvement of the fatty taste found in VSG-DIO mice.

Trp catabolism along the Kyn pathway is receiving an increasing attention in human because of its implication in the etiology of several diseases, such as cancers, irritable bowel syndrome, neuropsychiatric disorders, and diabetes [56]. The Kyn pathway is also known to be especially activated during human obesity [57]. To further explore the putative effects of the Kyn pathway on the fatty taste perception, an exploratory investigation was undertaken in patients in whom we have previously determined the LA perception thresholds before and after VSG [32]. This previous investigation had allowed to distinguish two subgroups: patients displaying an improvement of the LA sensitivity after surgery (i.e., lower thresholds) and not improved patients. A positive association between plasma Kyn levels and LA perception thresholds (i.e., high Kyn levels = low LA sensitivity) was only observed in the LA improved patients. Interestingly, this sub-group of patients displays a significant post-surgery reduction of its inflammatory status, as suggested by the decrease of plasma CRP level (Figure 5D). Therefore, it is tempting to speculate that this new inflammatory balance contributes to the reduction of the Kyn synthesis observed in LA-improved patients, the IDO-1 activity being inflammatory sensitive (Figure 2E).

To our knowledge, we bring herein the first evidence about a plausible involvement of the Trp/Kyn metabolism in the degradation of fatty taste perception generally associated with obesity. The major strength of this study is the pharmacological demonstration that the Kyn pathway inhibition is associated with an improvement of the peripheral gustatory function (increased Fun number) and of the perception of oily solution in the mouse. Nevertheless, whether these impacts are direct or indirect remains to be determined. Another limitation concerns the functional link between the Kyn pathway and the Fun papillae density which is based on associations rather than causal relationships. This question needs to be confirmed by mechanistic studies. Nevertheless, our data provide a possible mechanistic explanation to the relationship between variation in the Fun number and inflammatory status previously reported by Kaufman and collaborators [23,48]. It is noteworthy that the Trp metabolism also generates other biological active compounds, as serotonin and indole derivatives, that are potential regulators of the taste function. Indeed, serotonin contributes to the transmission of taste signals to the brain [58], while gut bacteria-derived indole metabolites affect the intestinal secretion of GLP-1 [59], known to modulate the fat taste sensitivity [60]. Moreover, it was recently reported that a Trp-derived indole metabolite reduced the weight gain in rats [61]. Collectively with the present study, these data emphasize the potential roles of Trp metabolites on both the taste function and obesity status, at least in rodents. Further experiments are required to fully explore this new paradigm.

In conclusion, this study strongly suggests that Trp/Kyn metabolism is one of regulatory factors in the cross-talk between nutritional obesity, bariatric surgery and taste of fat. A better understanding of molecular mechanisms responsible for obesity-mediated changes of the orosensory perception of energy-dense foods might lead to new nutritional and/or pharmacological strategies facilitating compliance with healthy dietary recommendations useful to avoid a weight rebound after bariatric surgery.

## Figures and Tables

**Figure 1 nutrients-13-01366-f001:**
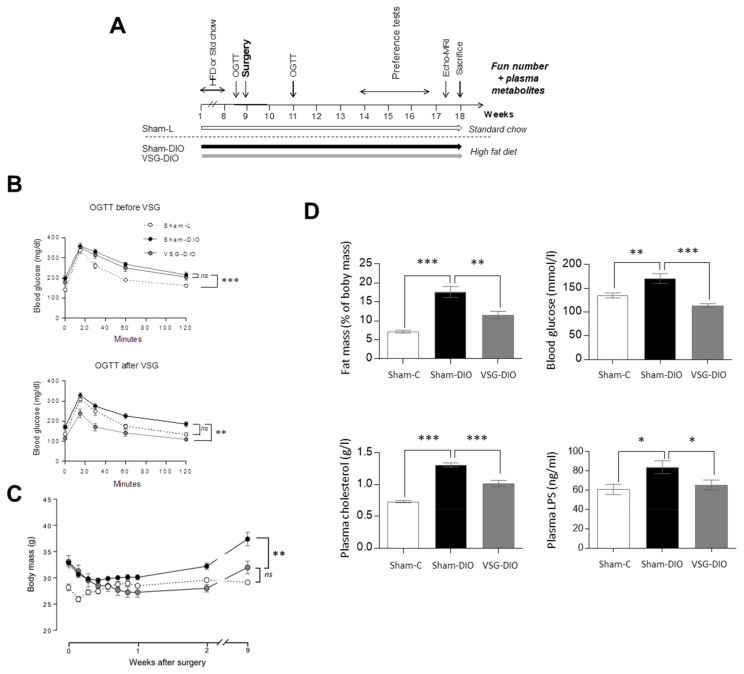
Comparison of body composition and of biochemical parameters in sham operated controls fed a standard chow (Sham-L, n=14) or a HFD (Sham-DIO, n=15) and in mice subjected to a vertical sleeve gastrectomy (VSG) fed the HDF (VSG-DIO, n=11). (**A**) Time-course of the experiment 1. (**B**) OGTT two days before then two weeks after surgery. (**C**) Evolution of body weight after surgery. (**D**) Fat mass, glycemia, plasma cholesterol and LPS levels. Mean ± SEM, Wilcoxon and Mann-Whitney tests: ns non-significant, * *p* < 0.05, ** *p* < 0.01, *** *p* < 0.001. DIO, diet induced obese; HFD, high fat diet; L, lean; LPS, lipopolysaccharides; OGTT, oral glucose tolerance test.

**Figure 2 nutrients-13-01366-f002:**
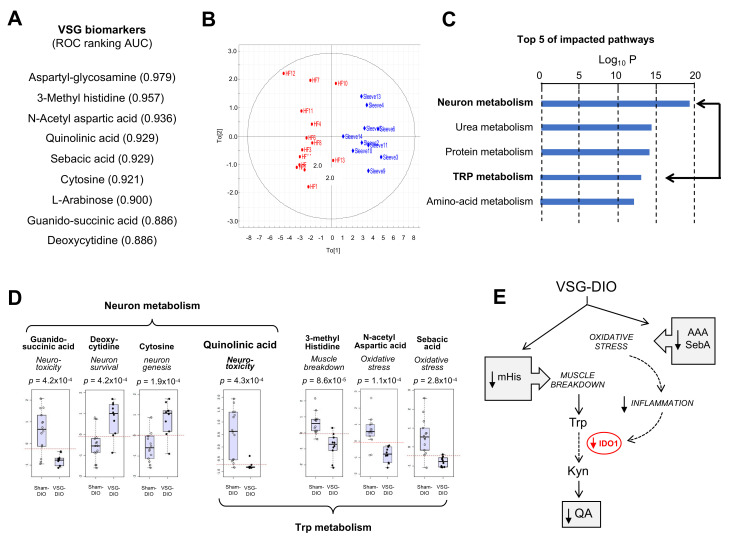
Metabolite biomarkers in plasma from the sham-operated obese mice (Sham-DIO, *n* = 15) and vertical sleeve gastrectomized obese mice (VSG-DIO, *n* = 11). (**A**) Identification of discriminant metabolites and ranking according to the biomarker performance using the receiver-operating characteristic (ROC) curves. (**B**) Principal component analysis. (**C**) Top Five of impacted metabolic pathways. (**D**) Identified plasma metabolites involved in neuronal function and tryptophan (Trp) metabolism. Boxplots derived from ROC analysis. (**E**) Simplified Trp catabolism along the kynurenine (Kyn) pathway. In grey, implication of identified metabolites. AAA, N-acetyl aspartic acid; IDO-1, indolamine 2,3 dioxygenase-1; mHis, methyl histidine; QA, quinolinic acid; SebA, sebacic acid.

**Figure 3 nutrients-13-01366-f003:**
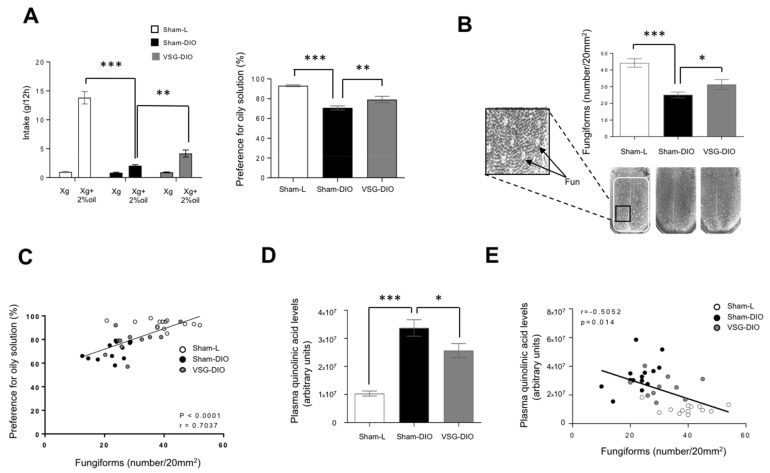
Behavioral phenotyping with respect to lipids in sham operated lean (Sham-L, *n* = 14) and obese (sham-DIO, *n* = 15) controls and in vertical sleeve gastrectomized obese mice (VSG-DIO, *n* = 11). (**A**) Long-term (12 h) two-bottle choice test. (**B**) Number of fungiform papillae (Fun) in the anterior part of the dorsal tongue. Mean ± SEM, Wilcoxon and Mann–Whitney tests: ** *p* < 0.01; *** *p* < 0.001. (**C**) Correlation between the preference for oily solution and the Fun number. Spearman correlation. (**D**) Plasma quinolinic acid levels at the sacrifice time. Mean ± SEM, Wilcoxon and Mann–Whitney tests: * *p* < 0.05; *** *p* < 0.001. (**E**) Correlation between plasma quinolinic acid and Fun number. Spearman correlation.

**Figure 4 nutrients-13-01366-f004:**
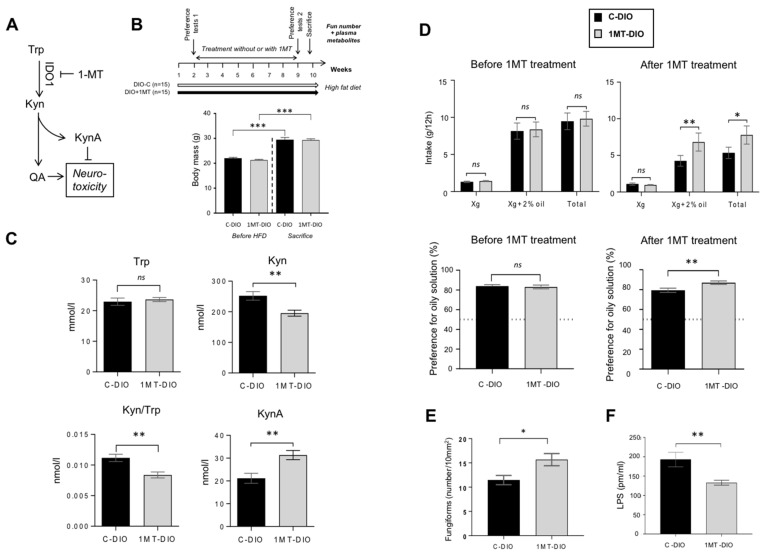
Pharmacological inhibition of the tryptophan (Trp) catabolism along the kynurenine (Kyn) pathway by the 1-methyl Trp (1-MT) inhibitor of rate-limiting enzyme indolamine 2,3 dioxygenase-1 (IDO-1). Obese mice received for 8 weeks 1-MT in drinking water (DIO + 1-MT, *n* = 15). Controls were DIO mice receiving water alone (DIO-C, *n* = 15). (**A**) Simplified Kyn pathway. (**B**) Time-course of the experiment and body mass before high fat diet (HFD) and at the sacrifice time. (**C**) Impact of the 1-MT treatment on plasma levels of key metabolites of the Trp catabolism. (**D**) Evolution of the preference for oily solution after 1-MT treatment. (**E**) Impact of the 1-MT treatment on the number of fungiform papillae (Fun). (**F**) Plasma lipopolysaccharides (LPS) levels. Mean ± SEM. Wilcoxon and Mann–Whitney tests: ns, non-significant; * *p* < 0.05; ** *p* < 0.01; *** *p* < 0.001. KynA, kynurenic acid; QA, quinolinic acid.

**Figure 5 nutrients-13-01366-f005:**
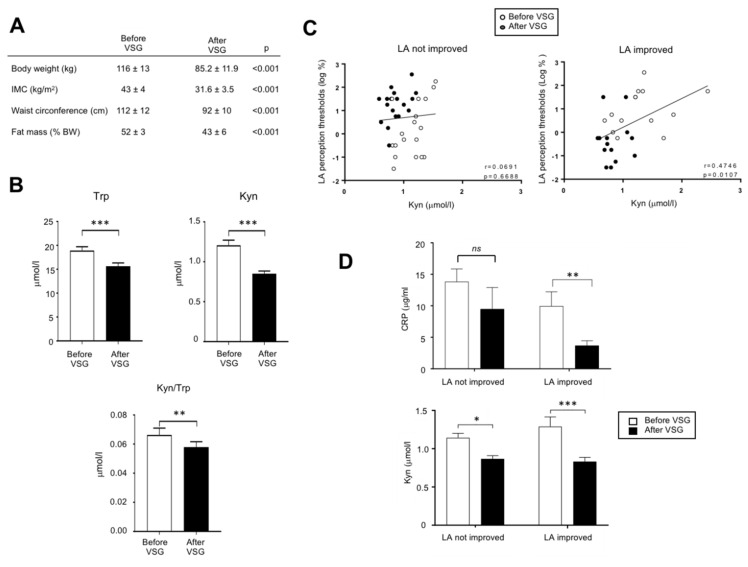
Analysis of the tryptophan metabolism in adult morbid obese female patients two weeks before and six months after vertical sleeve gastrectomy (VSG, *n* = 32). (**A**) Evolution of anthropometric parameters. (**B**) Effect of VSG on the plasma tryptophan (Trp) and kynurenine (Kyn) levels. (**C**) Correlations between the orosensory linoleic acid (LA) perception thresholds determined using the 3-ascending force-choice test and plasma Kyn levels in patients displaying an improvement of oral LA perception (i.e., reduction of LA perception threshold after VSG) and unresponsive patients. (**D**) Plasma C-reactive protein (CRP) levels. ns, non-significant; * *p* < 0.05; ** *p* < 0.01; *** *p* < 0.001.

## Data Availability

The data presented in this study are available on request from the corresponding author. The data are not publicly available due to ethical restrictions.

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
