# Peer review of "The Tryptophan/Kynurenine Pathway: A Novel Cross-Talk between Nutritional Obesity, Bariatric Surgery and Taste of Fat"

_nutrients, 2021, doi:10.3390/nu13041366_

Round 1
Reviewer 1 Report
nothing to say. If we agree with background the paper is coherent in all its section. to read the manuscript was difficult. Too much data in a little space, and sometimes it is not so easy to find a correspondence among the table and figure description and the text.
Author Response
Reviewer 1 :
nothing to say. If we agree with background the paper is coherent in all its section. to read the manuscript was difficult. Too much data in a little space, and sometimes it is not so easy to find a correspondence among the table and figure description and the text.
The manuscript has been improved to facilitate its reading, according to the reviewer’s recommendation

Reviewer 2 Report
Authors in the article entitled: "The Tryptophan/Kynurenine Pathway, a Novel Cross-Talk between Nutritional Obesity, Bariatric Surgery and Taste of Fat." decided to investigate correlations between fat intake, bariatric surgery and metabolism of Trp-Kyn pathway. The concept of this research project is interesting and in line with current research interest on associations between diet, weight gain and ways of possible treatment. This topic is especially important, because incidence of obesity still increases significantly.
Broad comments
Authors showed many interesting results from mice and humans experiments. This gives additional value to this project. Overall, it is a well thought and written article giving new insight into pathophysiology of obesity and sources for future interesting research project. Despite these positive aspects there are some important issues that has not been adressed in present work.
Specific comments
- This project included broad range of laboratory techniques to show different aspects of metabolism and fat taste. These techniques were properly explained in methodology section and clearly presented in the results.
- Authors focused mainy on kynurenic pathway, that is indeed major way of its metabolism. However, it is not the only option for synthesis of its metabolites. Tryptophan undergoes many biochemical reaction in mammals that lead to production of biologically active compounds. There are 3 main pathways, kynurenic, serotonin and indolic pathways that should be at least introduced in 1-2 sentences. Additionally, authors should explain in discussion why other pathways were not investigated.
- Authors decided to investigate IDO-1 inhibition in one part of experiments. Authors state that due to decrease in markers of inflammation was observed and positive impact on gustatory function. It was not investigated whether this inhibition increased metabolism of tryptophan by other pathways, that might later on reveal this effect. Authors; however, wrote "Nevertheless, whether these impacts are direct or indirect remains to be determined." it is satisfactory for the reviewer if in other section there will be explanation of other possible transformations of this amino acid.
- It is worth noting that tryptophan itself when administered decreases weight gain. It has been proven in multiple research projects. Additionally, recent discovery from 2019 showed that also indole-3-propionic acid, gut bacteria-derived metabolite of tryptophan reduces weight gain in rats. It might suggest that metabolites of tryptophan both endogenous and bacterial can regulate body mass and have positive metabolic impact. It would be worth noting in discussion section as a limitation or perspective for further research projects.
- Authors should recheck sentence in line 454-454. It was meant to be reduction or increase in this ratio? When you discuss a ratio it should be consistent throughout the manuscript. Blockage of IDO-1 I believe should decrease production of Kyn, and Trp levels will be higher then Kyn. It should be revised.
Author Response
Reviewer 2:
Authors in the article entitled: "The Tryptophan/Kynurenine Pathway, a Novel Cross-Talk between Nutritional Obesity, Bariatric Surgery and Taste of Fat." decided to investigate correlations between fat intake, bariatric surgery and metabolism of Trp-Kyn pathway. The concept of this research project is interesting and in line with current research interest on associations between diet, weight gain and ways of possible treatment. This topic is especially important, because incidence of obesity still increases significantly.
Broad comments
Authors showed many interesting results from mice and humans experiments. This gives additional value to this project. Overall, it is a well thought and written article giving new insight into pathophysiology of obesity and sources for future interesting research project. Despite these positive aspects there are some important issues that has not been adressed in present work.
Specific comments
- This project included broad range of laboratory techniques to show different aspects of metabolism and fat taste. These techniques were properly explained in methodology section and clearly presented in the results.
- Authors focused mainy on kynurenic pathway, that is indeed major way of its metabolism. However, it is not the only option for synthesis of its metabolites. Tryptophan undergoes many biochemical reaction in mammals that lead to production of biologically active compounds. There are 3 main pathways, kynurenic, serotonin and indolic pathways that should be at least introduced in 1-2 sentences. Additionally, authors should explain in discussion why other pathways were not investigated.
The following sentence was added in the Discussion: “It is noteworthy that the Trp metabolism also generates other biological active compounds, as serotonin and indole derivatives, that are potential regulators of the taste function. Indeed, serotonin contributes to the transmission of taste signals to the brain (Rittinger J J Biol Chem 279, 2004), while gut bacteria-derived indole metabolites affect the intestinal secretion of GLP-1 (Chimerel C Cell reports 2014), known to modulate the fat taste sensitivity (martin, JLR, 2012). Moreover, it was recently reported that a Trp-derive indole metabolite reduced the weight gain in rats (Konopelski, Nutrients 2019). Collectively with the present study, these data emphasize the potential roles of Trp metabolites on both the taste function and obesity status, at least in the rodents. Further experiments are required to fully explore this new paradigm”.
- Authors decided to investigate IDO-1 inhibition in one part of experiments. Authors state that due to decrease in markers of inflammation was observed and positive impact on gustatory function. It was not investigated whether this inhibition increased metabolism of tryptophan by other pathways, that might later on reveal this effect. Authors; however, wrote "Nevertheless, whether these impacts are direct or indirect remains to be determined." it is satisfactory for the reviewer if in other section there will be explanation of other possible transformations of this amino acid.
See above
- It is worth noting that tryptophan itself when administered decreases weight gain. It has been proven in multiple research projects. Additionally, recent discovery from 2019 showed that also indole-3-propionic acid, gut bacteria-derived metabolite of tryptophan reduces weight gain in rats. It might suggest that metabolites of tryptophan both endogenous and bacterial can regulate body mass and have positive metabolic impact. It would be worth noting in discussion section as a limitation or perspective for further research projects.
See above
- Authors should recheck sentence in line 454-454. It was meant to be reduction or increase in this ratio? When you discuss a ratio it should be consistent throughout the manuscript. Blockage of IDO-1 I believe should decrease production of Kyn, and Trp levels will be higher then Kyn. It should be revised.
The sentence has been modified according to this observation.

Reviewer 3 Report
The manuscript “The Tryptophan/Kynurenine Pathway, a Novel Cross-Talk between Nutritional Obesity, Bariatric Surgery and Taste of Fat” is a very interesting manuscript, presenting a study using animal models, for which some of the results were further validated in humans. This study show, for the first time, evidences of one pathway through which obesity (due to overconsumption of fat) affects fat taste.
The manuscript is well written and results clearly presented and discussed.
Even so, I have some minor points to be addressed, that I consider necessary to improve the clarity and presentation of the article.
Material and methods
2.1. Mice – Since information was collected from animals under different treatments, I suggest to add a scheme that allow to better follow all experimental procedure. Although Figure 1 contains one schematic representation of the different time-points of the experiment, I still miss a more clear scheme, namely presenting the different groups. After reading, I still am in doubt if I correctly understood the animals used: 70 mice in total? 11 obese with VSG; 14 lean (probably standard diet), 15 sham-DIO (for the first experiment) and 15 non-operated DIO treated with MT and 15 non-operated DIO not treated with MT (for the second experiment). But all this could benefit from the referred scheme.
Line 147 – “samples” – since these were centrifuged, these samples refer to supernatant or precipitate? Please specify.
Results
Lines 266-273 – This part has some discussion. I suggest the authors to keep the results presentation in this section and try to justify these results only in discussion section.
Lines 292-295 – I think this introduction is not necessary here. The authors refer that they want to test the potential role of kyn pathway, but the subsequent lines reports results obtained in animals without the inhibition of this pathway. I.e. the subsequent lines do not report results from the experiment made to test the role of kyn pathway.
Lines 312-316 – Once more, authors are discussing the results. Although I understand that sometimes it can be useful to present some tentatively justification for the observations, I suggest to keep this section only with results presentation.
Lines 337-339 – Avoid this discussion. Please see my previous comment.
Author Response
Reviewer 3:
The manuscript “The Tryptophan/Kynurenine Pathway, a Novel Cross-Talk between Nutritional Obesity, Bariatric Surgery and Taste of Fat” is a very interesting manuscript, presenting a study using animal models, for which some of the results were further validated in humans. This study show, for the first time, evidences of one pathway through which obesity (due to overconsumption of fat) affects fat taste.
The manuscript is well written and results clearly presented and discussed.
Even so, I have some minor points to be addressed, that I consider necessary to improve the clarity and presentation of the article.
Material and methods
2.1. Mice – Since information was collected from animals under different treatments, I suggest to add a scheme that allow to better follow all experimental procedure. Although Figure 1 contains one schematic representation of the different time-points of the experiment, I still miss a more clear scheme, namely presenting the different groups. After reading, I still am in doubt if I correctly understood the animals used: 70 mice in total? 11 obese with VSG; 14 lean (probably standard diet), 15 sham-DIO (for the first experiment) and 15 non-operated DIO treated with MT and 15 non-operated DIO not treated with MT (for the second experiment). But all this could benefit from the referred scheme.
According to the reviewer’s suggestion, in Fig. 1-A, the number of mice/group was added, the type of diets (std chow or HFD) was indicated for each group and in Fig. 4-B, the time-course of the &MT experiement was added.
Line 147 – “samples” – since these were centrifuged, these samples refer to supernatant or precipitate? Please specify.
Done
Results
Lines 266-273 – This part has some discussion. I suggest the authors to keep the results presentation in this section and try to justify these results only in discussion section.
We agree that this short paragraph, which introduces the identified discriminant metabolites (Fig. 2-D), can look like a discussion, especially due to its set of references. However, as this information can be useful for the reader, we have chosen to keep it in the Results section.
Lines 292-295 – I think this introduction is not necessary here. The authors refer that they want to test the potential role of kyn pathway, but the subsequent lines reports results obtained in animals without the inhibition of this pathway. I.e. the subsequent lines do not report results from the experiment made to test the role of kyn pathway.
The following sentence was suppressed “The fact that an overactivation of IDO-1 is found in obese mice (present data and [48]) has led us to explore the role that the Kyn pathway could play in the relationships between nutritional obesity and the perception of fatty taste and, consequently, in the improvements seen in VSG mice.”
Lines 312-316 – Once more, authors are discussing the results. Although I understand that sometimes it can be useful to present some tentatively justification for the observations, I suggest to keep this section only with results presentation.
The following senstence was suppressed “Collectively, these data suggest that a greater Trp catabolism along the Kyn pathway occurring in DIO mice might contribute to the degradation of their orosensory detection of oily stimuli”.
Lines 337-339 – Avoid this discussion. Please see my previous comment.
The following senstence was suppressed: “Daily water intake is tightly related to the body mass [52], DIO mice being characterized by a lower water consumption than lean controls [53].”
